# Simultaneous Expression of Different Therapeutic Genes by Infection with Multiple Oncolytic HSV-1 Vectors

**DOI:** 10.3390/biomedicines12071577

**Published:** 2024-07-16

**Authors:** Adriana Vitiello, Alberto Reale, Valeria Conciatori, Anna Vicco, Alfredo Garzino-Demo, Giorgio Palù, Cristina Parolin, Jens von Einem, Arianna Calistri

**Affiliations:** 1Department of Molecular Medicine, University of Padua, 35121 Padua, Italy; adriana.vitiello@unipd.it (A.V.); alberto.reale@unipd.it (A.R.); valeria.conciatori@studenti.unipd.it (V.C.); anna.vicco@studenti.unipd.it (A.V.); alfredo.garzinodemo@unipd.it (A.G.-D.); giorgio.palu@unipd.it (G.P.); cristina.parolin@unipd.it (C.P.); 2Department of Microbial Pathogenesis, School of Dentistry, University of Maryland, Baltimore, MD 21201, USA; 3Department of Microbiology and Immunology, School of Medicine, University of Maryland, Baltimore, MD 21201, USA; 4Institute for Virology, University of Ulm, 89081 Ulm, Germany

**Keywords:** virotherapy, immunotherapy, HSV-1, combinatorial approach

## Abstract

Oncolytic viruses (OVs) are anti-cancer therapeutics combining the selective killing of cancer cells with the triggering of an anti-tumoral immune response. The latter effect can be improved by arming OVs with immunomodulatory factors. Due to the heterogeneity of cancer and the tumor microenvironment, it is anticipated that strategies based on the co-expression of multiple therapeutic molecules that interfere with different features of the target malignancy will be more effective than mono-therapies. Here, we show that (i) the simultaneous expression of different proteins in triple-negative breast cancer (TNBC) cells can be achieved through their infection with a combination of OVs based on herpes simplex virus type 1 (oHSV1), each encoding a single transgene. (ii) The level of expressed proteins is dependent on the number of infectious viral particles utilized to challenge tumor cells. (iii) All recombinant viruses exhibited comparable efficacy in the killing of TNBC cells in single and multiple infections and showed similar kinetics of replication. Overall, our results suggest that a strategy based on co-infection with a panel of oHSV1s may represent a promising combinatorial therapeutic approach for TNBC, as well as for other types of solid tumors, that merits further investigation in more advanced in vitro and in vivo models.

## 1. Introduction

Oncolytic virotherapy is a promising anti-tumor treatment that lies at the intersection of cytolytic therapy, gene therapy, and immunotherapy [1]. Oncolytic viruses (OVs) selectively replicate in cancer cells, either naturally or through genetic manipulation [2], without causing relevant clinical disease. OVs are typically attenuated viruses with limited ability to counteract the intrinsic antiviral signaling pathways of healthy cells [3,4,5]. Their replication is favored in cancer cells because numerous antiviral pathways exert an antiproliferative effect and thus are compromised in malignancies [6,7]. To date, various OVs based on different viruses have been tested in clinical trials and have shown an excellent safety profile, but limited therapeutic efficacy [5,6,8]. It is widely accepted that OVs could become successful anti-cancer agents that combine selective killing activity with the triggering of an immune response against tumor cells. This goal could be achieved if viral replication elicits immunogenic cell death (ICD) along with the release of pathogen-associated molecular patterns (PAMPs) and damage-associated molecular patterns (DAMPs) in the tumor microenvironment (TME) [1,9]. As a consequence, the host immune response is boosted not only against viral antigens but also against tumor antigens [10,11]. Since OVs can be armed to express therapeutic transgenes, several studies evaluated factors, including cytokines and chemokines, that could enhance the immunogenicity of OVs [9,12]. However, it is well known that many solid tumors are surrounded by a TME with strong immunosuppressive properties, making it very difficult for the immune system to recognize and attack cancer cells [13]. A breakthrough in the treatment of some of these aggressive immunological cold solid tumors was provided by the development and implementation of immune checkpoint inhibitors (ICIs) [14]. Consequently, studies have explored the possibility of using OVs in conjunction with ICIs for the treatment of certain malignancies [15]. Indeed, the aforementioned immunotherapeutic mechanisms of OVs may increase inflammation and recruitment of T lymphocytes to the tumor bed, thereby sensitizing the tumor mass to ICI treatment [16].

The first OV approved for clinical use in the United States and Europe was Talimogene Laherparepvec or T-VEC (Imlygic, Amgen), which is used for the intralesional treatment of advanced, surgically unresectable melanoma [11,17,18]. T-VEC is an oncolytic herpes simplex type 1 virus (oHSV-1) containing a deletion of Us12 and both γ34.5 genes [19]. The γ34.5 gene encodes for ICP34.5, a well-known HSV-1 neurovirulence factor that is also involved in viral escape from the innate immune response of host cells [20]. Therefore, its deletion strongly attenuates HSV-1, as in vivo studies have shown [8], and additionally limits viral replication selectively to cells that exhibit defects in the interferon signaling pathways, a typical feature of cancer cells [21]. Due to the absence of ICP47, the protein encoded by Us12, T-VEC-infected cells also efficiently present antigens and thus enhance the anti-tumor immune response [11]. Finally, T-VEC is armed with the human granulocyte–monocyte colony-stimulating factor (GM-CSF), a cytokine that stimulates dendritic cells [19]. The effectiveness of T-VEC in the treatment of melanoma is not unexpected, as this tumor is known to be highly immunogenic and often responds to other forms of immunotherapy, including ICIs [22]. However, other types of solid tumors are surrounded by a highly immunosuppressive TME, making it very difficult for the immune system to recognize and attack cancer cells, even after ICI and/or OV treatment [5,14]. Another obstacle to effective in vivo viro-immunotherapy is the inter- and intra-tumoral heterogeneity of cancer cells and of the immunosuppressive mechanisms of the TME [23]. This heterogeneity suggests that combining different therapeutic interventions may be necessary to achieve an effective strategy for the treatment of most malignancies [1,5,9].

Here, we show the simultaneous expression of reporter (enhanced green fluorescent protein [EGFP], firefly luciferase [Fluc]) and immunotherapeutic (human interleukin 12 [IL12]) proteins in triple-negative breast cancer (TNBC) cells after their infection with a combination of oHSV-1s sharing the Δγ34.5/ΔUs12 backbone. Overall, our data suggest that the expression levels of the transgenes can be tuned by adjusting the viral inoculum used for co-infections. Importantly, the effects of each recombinant oHSV-1 on the viability of the cancer cells and viral replication kinetics were comparable in mono-infected and co-infected cells. These findings indicate that the simultaneous administration of more than one OV, each carrying a specific transgene, results in the simultaneous expression of multiple factors, while maintaining the ability of the recombinant viruses to kill tumor cells. This combinatorial strategy could thus enable a therapeutic approach that can be easily tailored to the specific characteristics of the cancer/patient to be treated.

## 2. Materials and Methods

Cells. Vero (green monkey kidney cells, obtained from the American Type Culture Collection [ATCC], LGC Standards s.r.l., Sesto San Giovanni, Italy, with reference # CCL-81T M), 293T (human embryonic kidney cells, ATCC, # CRL-3216), and MDA-MB-231 (triple-negative human breast cancer cells, ATCC, # HTB-26) cell lines were maintained in Dulbecco’s modified eagle medium (DMEM, Gibco, ThermoFisher Scientific, Rodano, Italy), supplemented with 1% *v*/*v* Penicillin–Streptomycin (Gibco) and 10% *v*/*v* Fetal Calf Serum (FCS, Gibco). Cells were split twice a week and cultured at 37 °C in a 5% CO_2_ atmosphere. To ensure the absence of mycoplasma contamination, cell lines were subjected to regular testing via end-point PCR using the AmpliTaq Gold™ DNA polymerase (ThermoFisher Scientific, Rodano, Italy), the forward 5′-GGGAGCAAACAGGATTAGATACCCT primer, and the reverse 5′-TGCAC-CATCTGTCACTCTGTTAACCTC primer.

Viruses and BAC mutagenesis. HSV-1 strain 17syn+ [24] was used as the wild-type virus. Viral stocks were prepared after cultivation in Vero cells, as described below. A bacterial artificial chromosome (BAC) containing the entire genome of HSV-1 strain 17syn+ with the deletion of both copies of the γ34.5 gene and the insertion of a cassette expressing firefly luciferase (Fluc) within the intergenic region of the viral open reading frames UL55 and UL56 (hereafter named BAC-Δγ34.5) was kindly provided by Prof. Beate Sodeik (University of Hannover) [25]. Δγ34.5/ΔUs12/Fluc HSV-1 and Δγ34.5/ΔUs12/EGFP HSV-1 (hereafter named ΔΔ-Fluc and ΔΔ-EGFP, respectively) were generated by BAC mutagenesis [26,27], and are described elsewhere [27]. Briefly, mutagenesis was performed using BAC-Δγ34.5 to achieve the same Us12 deletion as described in T-VEC [19], generating ΔΔ-Fluc. It has already been shown that the insertion of an expression cassette into the viral intergenic region UL55-UL56 enables a durable expression of the transgene in vitro and in vivo [25]. For the generation of ΔΔ-EGFP, the Fluc-coding sequence was therefore replaced by a sequence coding for enhanced green fluorescent protein (EGFP), so that EGFP is expressed under the control of the same promoter as Fluc in ΔΔ-Fluc, the immediate early cytomegalovirus promoter (P_CMV_). A similar strategy was used to replace the Fluc-coding sequence with the sequences coding for human interleukin 12 (IL12). The two subunits (p35 and p40) that constitute the functional protein were obtained from the pCMVIL-12neo plasmid, kindly provided by Dr. Egilmez (Roswell Park Cancer Institute, Buffalo, New York 14263, USA). In the ΔΔ-IL12 construct, p35 and p40 are separated by an Internal Ribosome Entry Site (IRES) to allow their simultaneous expression [28]. DNAs of all generated bacmids (ΔΔ-Fluc, ΔΔ-EGFP, ΔΔ-IL12) were used to transfect 293T cells, and viral stocks were produced by amplification and titration of recombinant viruses in Vero cells, as described below.

Reconstitution, amplification, and titration of recombinant viruses. To reconstitute the recombinant viruses, the respective bacmid DNAs were used to transfect 293T cells using Lipofectamine 2000™ (Invitrogen, ThermoFisher Scientific), following manufacturer’s instructions. Upon transfection, cells were monitored for viral cytopathic effect and when possible, for transgene expression, e.g., EGFP fluorescence. When cytopathic effects were observed in roughly 90% of the cell monolayer (typically after 72 h), cell culture supernatants were harvested and used to infect Vero cells. Once again, cells were monitored for cytopathic effects and transgene expression. Virus stocks were generated after cultivation for 48–72 h at 37 °C by harvesting infected cells and subjecting them to 3× freeze-and-thaw cycles. Next, cellular debris was removed by centrifugation at 1000 rpm for 15 min at 4 °C, and the supernatant was aliquoted and stored at −80 °C until use. The same protocol was employed to harvest supernatant for HSV-1 strain 17syn+ or any other viral stock, by employing a multiplicity of infection (MOI) of 0.01 plaque-forming units (PFU) per cell for the infection of Vero cells.

Infectious virus titers of viral stocks were determined by titration on Vero cells. Briefly, 10-fold dilutions of viral stocks were used to infect Vero cells (10^5^ cells/well) in 24-well plates at 37 °C for 1 h. Cells were then washed three times with phosphate-buffered saline (PBS), and overlayed with 2% *w*/*v* carboxymethyl-cellulose in 2% *v*/*v* FBS DMEM for 72 h. Next, cells were fixed with 5% *v*/*v* formalin and stained with 0.5% *w*/*v* crystal violet in 25% *v*/*v* methanol. All the reagents were purchased from Merck Life Science (Milan, Italy). Viral titers of viral stocks were calculated by counting plaques and expressed as PFU/mL. The same protocol was used to titrate infectious viral particles released in the cell supernatants of breast cancer cell lines infected with recombinant viruses, as reported below.

The yield of infectious particles of each viral stock was further confirmed by infecting Vero cells with an MOI of 0.01 PFU/cell, based on the calculated titers. Six hours post infection (hpi), cells were fixed and the expression of the viral immediate early protein ICP4 was detected by indirect immunofluorescence staining, following the protocol reported below. ICP4-positive cells were then counted to confirm an infection rate of MOI 0.01. 

Finally, the stability of recombinant viruses was controlled by their continuous passage in Vero cells for at least 28 days. For this purpose, Vero cells were cultured in 75 cm^2^ flasks and infected with ΔΔ-Fluc, ΔΔ-EGFP, and ΔΔ-IL12, respectively, at an MOI of 0.01 PFU/cell. Seventy-two hours post infection, cell supernatants were harvested and used to infect Vero cells in a new 75 cm^2^ flask. This procedure was repeated a total of 8 times. Virus growth was monitored regularly using a Leica epifluorescence DC100 microscope by observing the presence of cytopathic effects (bright fields). When possible, the expression of transgenes was verified, e.g., EGFP expression with a fluorescence microscope (Leica Microsystems, Varese, Italy).

Infection of breast cancer cell line. MDA-MB 231 cells were seeded in 24-well plates (1.8 × 10^5^ cells/well), 6-well plates (8.7 × 10^5^ cell/well), or T25 flasks (2.3 × 10^6^ cells/well) depending on the aim of the experiment. The day after, cells were infected with recombinant viruses at the appropriate MOI in serum-free DMEM for 1 h at 37 °C. Next, cells were washed 3× with PBS and maintained in 2% FBS DMEM. At different times post infection, cell supernatants were harvested, and virus yields were determined by titration on Vero cells. From the infections of MDA-MB-231 cells, either the cell lysates or cell supernatants were collected at 72 hpi and subjected to the appropriate assay for transgene expression analysis as described below.

Analysis of transgene expression. The expression of EGFP was examined by imaging the infected cells with a fluorescence microscope (Leica). In addition, EGFP fluorescence and Fluc activity were measured in cell lysates using a multi-label plate reader (VICTOR X2, PerkinElmer, Waltham, MA, USA), as previously described [29].

IL12 production and release were determined in the supernatants of infected cells by a human IL-12 p70 ELISA kit (Invitrogen, ThermoFisher Scientific) according to the manufacturer’s instructions. Prior to measurement, proteins from the cell supernatant were concentrated by centrifugation at 8000 rpm for 15 min using Vivaspin^®^ 2 centrifugal concentrator polyethersulfone columns (MWCO 10 kDa, Sortorius Italy s.r.l., Varedo, Italy).

Immunofluorescence assay. In the experiments comparing the virus spread, killing activity, and transgene expression of different viruses, the same MOI was used for a comparable infection of MDA-MB-231 cells, based on the virus titer of the virus stocks. Furthermore, to ensure comparable infection, the expression of the HSV-1 immediate early protein ICP4 was controlled via immunofluorescence at 6 hpi. For this purpose, MDA-MB-231 cells were seeded in 24-well plates at a density of 1.8 × 10^5^ cells per well on round coverslips. Twenty-four hours later, the cells were infected with the appropriate recombinant HSV-1, at the desired MOI, in serum-free DMEM for 1 h at 37 °C. Six hours post infection, cells were fixed and permeabilized with 100% methanol for 10 min at −20 °C, washed with PBS, and incubated overnight at 4 °C with primary anti-ICP4 antibody (10F1, Abcam, Prodotti Gianni, Milan, Italy) diluted 1:100 in BSA 2.5% *v*/*v* in PBS. The next day, the samples were incubated for 1 h at room temperature with a secondary goat anti-mouse antibody (Alexa Fluor^®^ 585, Abcam, Prodotti Gianni, Milan, Italy). The cell nuclei were stained with DRAQ5™ (ThermoFisher Scientific). The samples were then analyzed using a confocal microscope (Stellaris, Leica Microsystems, Varese, Italy).

The same test was used on infected cells at 72 hpi to analyze virus spread.

## 3. Results

Deletion of the γ34.5 gene encoding ICP34.5 is a cornerstone in the development of oncolytic HSV-1 (oHSV-1) vectors, as ICP34.5 is an important virulence factor in vivo but largely dispensable in cell culture. In search for oHSV-1 candidates against human breast cancer, we evaluated the replication capacity of a Δγ34.5 HSV-1 in the MDA-MB-231 cell line, a widely accepted cell culture model for triple-negative breast cancer (TNBC) [30]. We decided to focus on TNBC as a model, as this is an aggressive tumor that responds poorly to current treatments and for which oncolytic viruses (OVs) may represent a therapeutic option, particularly in synergy with immunotherapy.

Initially, we compared the replication and killing activities of wild-type HSV-1 (laboratory strain 17+) to those of a Δγ34.5 derivative in the MDA-MB-231 cell line. Cells were infected with the same number of infectious viral particles [multiplicity of infection (MOI) of 0.01 plaque-forming units (PFU)/cell], resulting in an initially similar infection. Indeed, detection of the HSV-1 immediate early protein ICP4 6 h post infection (hpi) demonstrated a comparable number of infected cells for both viruses (Figure 1a, ICP4 and merge panels at 6 hpi). However, wild-type HSV-1 (17+) grew more efficiently over time than the Δγ34.5 derivative, a finding which was also reflected by differences in extracellular virus yields (Figure 1b and Appendix A). In addition, 17+ caused almost complete destruction of the cell monolayer at 72 hpi (Figure 1a, panels at 72 hpi), which is confirmed by the low percentage of viable cells compared to uninfected samples and Δγ34.5 infection (Figure 1c).

It was previously reported that defects in Δγ34.5 HSV-1 replication in certain cancer cells are overcome by the additional deletion of the Us12 gene [11,19,30]. Therefore, we took advantage of a mutant virus, hereafter named ΔΔ-Fluc (Figure 2a), that we previously generated [26]. ΔΔ-Fluc is a Δγ34.5 HSV-1 into which the same Us12 deletion that characterizes T-VEC [19] was introduced using bacterial artificial chromosome (BAC) mutagenesis. Moreover, as with the original bacmid [23], ΔΔ-Fluc contains the firefly luciferase (Fluc) gene under the transcriptional control of the human cytomegalovirus immediate early promoter (P_CMV_), which has been inserted in the UL55-UL56 intergenic region of the virus. As expected, ΔΔ-Fluc replicated to higher extracellular virus yields (Figure 2b and Appendix A) and had a significantly higher impact on cell viability in infections of MDB-MB-231 cells using an MOI of 0.1 PFU/cell (Figure 2c) compared to Δγ34.5.

Encouraged by these results, we substituted the sequence encoding for Fluc with the gene expressing human interleukin 12, generating the recombinant virus ΔΔ-IL12 (Figure 3a). Human interleukin 12 (IL12) is a dimeric cytokine known to activate both adaptive and innate immunity. IL12 was previously used to arm oncolytic viruses to enhance their ability to elicit an immune response against tumor cells [9,31,32]. We also tested ΔΔ-EGFP, a recombinant HSV-1 that we had previously developed [26], in which EGFP was cloned in place of Fluc. ΔΔ-EGFP allows for easy visualization of transgene expression in infected cells, which has already been shown to be stable in vitro in cell lines of different origin and in vivo in the chicken embryo chorioallantoic membrane model [26]. The growth of ΔΔ-IL12 and ΔΔ-EGFP was analyzed in MDA-MB-231 cells infected at an MOI of 0.1 PFU/cell. At 72 hpi, ΔΔ-EGFP and ΔΔ-IL12 exhibited comparable spreading infections in MDA-MB-231 cells (Figure 3b), and both viruses demonstrated a similar impact on cell viability, which was comparable to that of the parental ΔΔ-Fluc virus (Figure 3c). It is noteworthy that transgene expression was detected in MDA-MB-231 cells, as evidenced by microscopy and the measurement of the fluorescence signal intensity in ΔΔ-EGFP-infected cells (Figure 3b, ΔΔ-EGFP panels, and Figure 3e), by a luciferase assay in ΔΔ-Fluc-infected cells (Figure 3d), and by an IL12 ELISA assay performed with the supernatants of ΔΔ-IL12-infected cells (Figure 3f).

To determine whether co-infection of cancer cells with different oHSV-1s could result in the simultaneous expression of several exogenous genes, we infected MDA-MB-231 cells with a combination of ΔΔ-EGFP and ΔΔ-Fluc at an MOI of 0.1 PFU/cell or of 0.05 PFU/cell for each virus. Additionally, cells were infected with single recombinant viruses (MOI 0.1 PFU/cell) to serve as a control. Brightfield and fluorescence microscopy revealed progressive cytopathic effects over the 72 h observation period (Figure 4a), with measurable effects on cell viability (Figure 4b). The intensity of Fluc and EGFP signals was also assessed (Figure 4c,d, respectively). These analyses showed that both EGFP and Fluc are expressed in co-infected MDA-MB-231 cells at both MOIs, albeit to a lesser extent than in single infections. Interestingly, the lowest transgene expression was observed at the highest MOI (0.1 PFU/cell for each virus), suggesting a stronger overall effect on cells and thus on transgene expression.

When Vero cells were infected with the supernatants harvested from mono- and co-infected MDA-MB-231 cells, expression of EGFP and Fluc was detectable in all cases (Appendix A). This result demonstrates that mixed infections of MDA-MB-231 cells produce infectious viral particles that are capable of expressing the respective transgenes in subsequent infections of another cell line.

Having demonstrated the co-expression of different reporter genes in MDA-MB-231 cells, we tested whether human IL12 could also be efficiently expressed together with a reporter gene via co-infection with a combination of different virus constructs. MDA-MB-231 cells were infected with the ΔΔ-IL12 construct in combination with a ΔΔ-EGFP construct under the same experimental conditions as described above. As observed through brightfield and fluorescence microscopy, a progressive cytopathic effect (rounding and detachment of cells) was evident during the 72 h observation period, accompanied by a clear decrease in cell viability at both MOIs utilized (Figure 5a,b). The intensity of the EGFP signal was also quantified, as was the amount of IL12 released in the cell supernatants (Figure 5c,d, respectively). Like the previous co-challenge experiments (Figure 4c,d), lower expression levels of both proteins were observed in the combinatorial infections, independent of the MOI used. In particular, IL12 was under the detection limit at an MOI of 0.1 PFU/cell, whereas it was measurable at an MOI of 0.05 PFU/cell, which confirms the results of the ΔΔ-EGFP/ΔΔ-Fluc mixed infections (Figure 5c).

Finally, we investigated the viral replication kinetics of co-infections compared to mono-infections. To this end, supernatants were harvested at different times after infection of MDA-MB-231 cells that were mono-infected or infected with a combination of ΔΔ-EGFP and either ΔΔ-Fluc or ΔΔ-IL12. Infectious virus yields in the supernatants were then determined by titration in Vero cells. Fluorescence microscopy was used to distinguish between EGFP-positive plaques and non-fluorescent plaques during the titration. The green plaques were counted as ΔΔ-EGFP plaques, while EGFP-negative plaques were counted as either ΔΔ-Fluc- or ΔΔ-IL12-positive plaques. All recombinant viruses replicated when used in combination with MDA-MD-231 cells with titers comparable to those achieved in single infections, without significant differences between the two MOIs used in combinatorial infections (Figure 6a,b).

## 4. Discussion

Virotherapy is emerging as an effective strategy for the treatment of solid tumors, especially if poorly responsive to currently available therapeutic approaches. In particular, the use of oncolytic viruses (OVs) is effective when their ability to kill cancer cells is combined with the triggering of an immune response against the tumor [1,2,3,4,5,9,33]. OVs can elicit an immune response per se, by simply infecting cancer cells. However, this property can be enhanced by combining OVs with other treatments or by arming the viruses with immune-therapeutic factors [5,9]. Given the heterogeneity of cancer, targeting multiple immunosuppressive mechanisms is crucial for efficacy and constitutes the basis for patient-tailored therapy [34]. To date, the HSV-1-based OV, T-VEC, is the most commonly used recombinant virus for cancer therapy. T-VEC was approved in 2015 by the Food and Drug Administration (FDA) and later also by the European Medicines Agency (EMA) for the treatment of patients with advanced melanoma. T-VEC is characterized by the expression of the granulocyte–monocyte colony-stimulating factor (GM-CSF) and the deletion of the ICP34.5- and ICP47-encoding genes. Overall, these modifications increase the preferential lytic viral replication in tumor cells, the safety of T-VEC, and its ability to elicit anti-tumoral immunity [11]. Indeed, T-VEC displays both a direct effect on cancer cells when administered intratumorally in vivo, and an immunological mechanism of indirect action that leads to the regression of untreated metastatic sites [17]. Current clinical experience data from patients who received T-VEC for cancers other than melanoma indicate that this OV is generally safe and causes only minor side effects [8]. However, some issues remain to be addressed, such as (i) how to improve its therapeutic efficacy, especially towards immunological cold tumors, and (ii) how to deliver it to difficult-to-access cancers, with efforts to develop strategies for systemic administration of T-VEC and other OVs [26,35].

Regarding the first point, viruses with a large DNA genome, such as HSV-1, allow the insertion of multiple therapeutic genes [5,36], thus providing useful tools for the development of combinatorial platforms. In this context, some studies have already reported successes in arming oncolytic HSV-1 (oHSV-1) to express a panel of therapeutic factors that complement the viral killing activity with the ability to target various features of tumors cells and TME [5,9,34,36]. However, this strategy suffers from some drawbacks, such as the difficulty of (i) modulating the transcription of the various transgenes in vivo, and (ii) rapidly tailoring the panel of factors to the characteristics of the cancer and its TME. Further, manipulation of the HSV-1 genome is time-consuming and costly.

The administration of a combination of oHSV-1s, each expressing a different therapeutic gene, could mitigate the above issues and offer certain additional advantages. Desired OVs could simply be selected from a pre-established library and mixed ad hoc to adjust the therapy to a particular type of tumor/TME or to address the needs of the individual patient. Combining such pre-engineered OVs is expected to be faster, more cost-effective, and more versatile than the ad hoc generation of new multigenic oHSV-1s for each individual application. In support of this concept, Xie et al. [37] showed that two oHSV1s, one expressing a mouse anti-PD-1 antibody (aPD1) and the other encoding for the murine IL-12, inhibited the growth of primary tumors when injected together in a colon adenocarcinoma mouse model. They also reported the induction of an anti-tumor vaccinal effect with a significant increase in the overall survival rate of mice [37]. The latter result is particularly interesting because the ability of OVs to elicit a systemic immune response against cancer cells is closely linked to the success of oncolytic virotherapy in clinical practice [1,3,5,9,10,12,15], e.g., the use of T-VEC in patients with melanoma [11,17,18]. Indeed, the effect of the virus on distant metastasis and on recurrent tumors is one of the unsolved problems with cancer therapy based on OV. Therefore, a vaccine-like response would be a clear advantage, supporting the strategy of arming OVs with immunotherapeutic molecules [9], such as IL12 [37], as we tested here. However, some potential drawbacks of this approach should be considered, such as the possibility that oHSV-1-based OVs armed with different transgenes may compete for target cells, which could lead to the rapid disappearance of one of the therapeutic factors from the tumor mass.

Our study aimed to address some of these issues in vitro using the MD-MBA-231 cell model for TNBC [30] and a series of viruses that either express reporter genes (EGFP or Fluc) or a candidate therapeutic factor (human IL12). We decided to focus on TNBC as this tumor represents a potential target of combinatorial immunovirotherapy. Indeed, trastuzumab and hormone-based therapies are ineffective in the treatment of this subtype of breast cancer, which has a poor prognosis and aggressive clinical behavior [38,39]. Furthermore, TNBC is characterized by a wide range of genetic and immunophenotypic features that would require a readily adaptable treatment [40]. Specifically, we investigated whether the use of multiple HSV-1-based OVs would allow (i) the simultaneous expression of each molecule and (ii) the regulated expression of transgenes by adjusting the viral inputs. We showed the following: 1.The deletion of the virulence gene γ34.5, which is known to attenuate HSV-1, abrogated the efficient killing of TNBC cells. This finding is not surprising, as it has already been reported that certain types of cancer cells are resistant to Δγ34.5 HSV-1 replication [19]. However, the limitation of virus growth due to the deletion of γ34.5 was compensated for by deleting the Us12 gene, analogous to the genetic modifications of T-VEC [19]. Deletion of Us12 results in altered expression kinetics of the Us11 gene, which enhances viral replication without compromising safety [13,18]. Indeed, a recombinant virus with a Δγ34.5/ΔUs12 genomic backbone (ΔΔ-Fluc) was more efficient in killing MD-MBA-231 cancer cells than a Δγ34.5 HSV-1.2.Various oHSV-1s based on the Δγ34.5/ΔUs12 backbone, alone or in combination, infect and kill MDA-MBA-231 cells with comparable efficiency, regardless of the expressed transgene.3.The transgenes are expressed in both mono- and co-infections. This result is consistent with the feasibility of a combinatorial strategy based on the simultaneous administration of individual oHSV-1s encoding for different therapeutic factors [34].4.The expression levels of each transgene were lower when the respective oHSV-1 was used in mixed infections than in mono-infections. This finding could be partially explained by promoter competition, since all transgenes are under the transcriptional control of the P_CMV_. Furthermore, we noticed an overall decrease in expression in co-infections when the cells were infected with higher viral inputs (MOI = 0.1 PFU/cell for each recombinant virus). This result probably reflects the effects of infection on the overall cellular metabolism. In particular, the amount of IL12, the therapeutic protein, was reduced in the combinatorial infections. As mentioned above, IL12 is composed of two subunits, whose coding sequences were cloned in the viral genome under the transcriptional control of P_CMV_ and separated by an internal ribosomal entry site (IRES). It can be expected that its translation process is less efficient than that of small proteins with only one subunit, such as EGFP or Fluc. These aspects should be carefully considered when it comes to finding the best combination of therapeutic molecules (and thus of specific oHSV-1s) and the respective viral inputs in vitro before moving on to in vivo challenges. On the other hand, the finding that the selected MOIs might affect transgene expression could have important translational value. Specifically, the expression of therapeutic factors could be “tuned” by adjusting the particle load of each oHSV-1 in the in vivo challenge, instead of genetic manipulations of promoters, and so on.5.All recombinant viruses showed comparable replication kinetics and comparable killing capacities in the cell lines tested here. The various genetic elements we inserted had no effect on viral fitness and did not cause any obvious imbalance in the replication of the oHSV-1s in co-infections. Whether these findings hold true for all transgenes remains to be determined empirically. Nevertheless, our results suggest that a combination of oHSV-1s could be a viable approach for expressing different transgenes without significant drawbacks in viral spread or killing efficacy. Conversely, the data indicate that the observed differences in transgene expression levels are not attributable to differences in infectivity or replication capacity of the oHSV-1 viruses, at least not over the 72 h observation period.

## 5. Conclusions

This proof-of-concept study, performed on a cell line that is widely used as an in vitro model of TNBC [30], supports the concept that a combination of oHSV-1s expressing different foreign genes is a feasible approach to delivering customized arrays of factors to cancer cells, while retaining viral replication and intrinsic killing activity against infected cells. These findings add up to the possibility of modulating the expression of different transgenes by optimizing viral inputs more than by working on the transcriptional control. Based on these encouraging results, this strategy should be further investigated in more complex in vitro (3D cultures) and in vivo cancer models to better assess the stability of the recombinant viruses over time, especially in the case of combined infections, their safety, and their ability to elicit an anti-tumoral immune response.

## Figures and Tables

**Figure 1 biomedicines-12-01577-f001:**
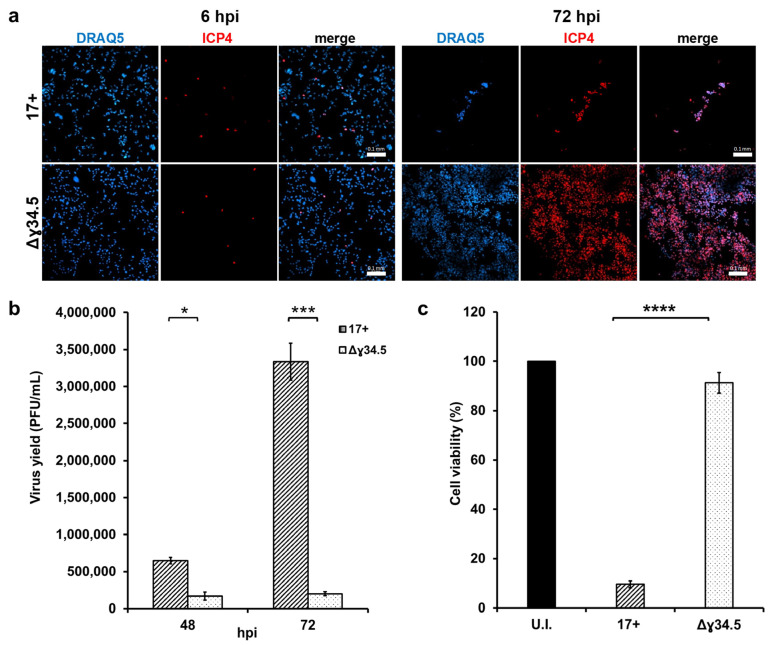
Δγ34.5 HSV-1 replicates in and kills TNBC cells less efficiently than wild-type HSV-1. MDA-MB-231 cells were infected with wild-type (17+) and γ34.5-deleted (Δγ34.5) HSV-1 at an MOI of 0.01 PFU/cell. (**a**) After 6 and 72 hpi, infected cells were subjected to indirect immunofluorescence staining for ICP4 by confocal microscopy. Nuclei were stained with DRAQ5. Representative images are reported. Scale bars: 0.1 mm. (**b**) Cell supernatants were collected at the indicated times and infectious viral particles were determined by titration on Vero cells. The graph shows the mean viral titer in PFU/mL and standard error (SE) from at least three independent experiments. (**c**) Cell viability was determined by trypan blue staining at 72 hpi. Bars display the mean value of the percentage of live cells from infections compared to uninfected cells from three independent experiments. Error bars represent SE. A Student *t* test was used for statistical analysis (* *p* value < 0.05; *** *p* < 0.001; **** *p* < 0.0001).

**Figure 2 biomedicines-12-01577-f002:**
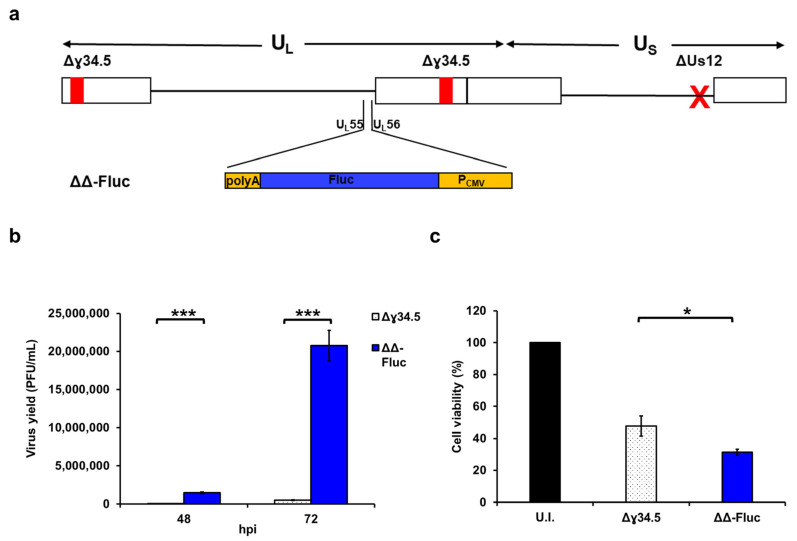
ΔΔ-Fluc replicates in MDA-MB-231 cells and kills them more efficiently than Δγ34.5 HSV-1. (**a**) Schematic of the bacmid of recombinant virus ΔΔ-Fluc. This construct is deleted of the two ɣ34.5 loci as well as of the Us12 gene and expresses Fluc under the control of the human cytomegalovirus immediate early (IE) promoter (P_CMV_). The P_CMV_/Fluc cassette is inserted into the UL55 and UL56 intergenic region of HSV-1. The red “X” indicates the Us12 deletion. (**b**) MDA-MB-231 cells were infected at an MOI of 0.1 PFU/cell. The graph displays the amounts of infectious virions released in the cell supernatants at indicated times post infection. Bars show the mean values and error bars from titrations performed in triplicate. (**c**) Cell viability was determined by trypan blue staining at 72 hpi. The graph shows the mean values of the percentage of live cells from infections compared to the uninfected cells from three independent experiments. Error bars represent the SE. A Student *t* test was used for statistical analysis (* *p* value < 0.05; *** *p* < 0.001).

**Figure 3 biomedicines-12-01577-f003:**
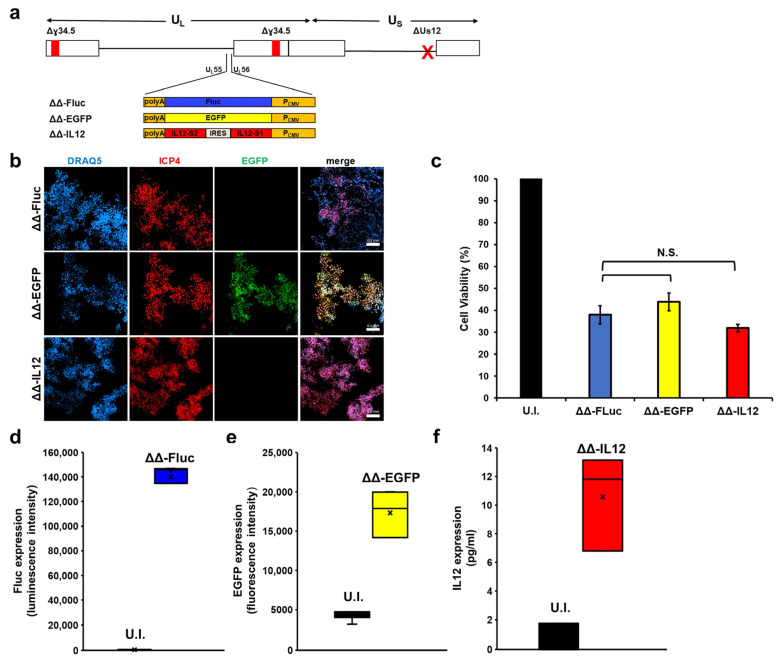
Expression of different transgenes from recombinant viruses in MDA-MB-231 cells. (**a**) Schematic of the bacmids expressing Fluc, EGFP, and IL12. IL12 S1 and IL12-S2 stand for IL12 subunit 1 (p35) and IL12 subunit 2 (p40), respectively. The red “X” indicates the Us12 deletion (**b**) MDA-MD-231 cells were infected with the indicated recombinant viruses (MOI = 0.1 PFU/cell). At 72 hpi, cells were stained with an ICP4-specific antibody. After incubation with DRAQ5 to mark nuclei, cells were observed by confocal microscopy. Representative images are shown. Scale bars: 0.1 mm. (**c**) Cells were harvested at 72 hpi and cell viability was evaluated by trypan blue staining. The graphs display the mean values of the percentage of dead cells compared to uninfected cells from at least three independent experiments. Error bars represent the SE. A Student *t* test was used for statistical analysis (N.S.: not statistically significant; *p* > 0.05). The expression of Fluc (**d**) and EGFP (**e**) was measured using a multi-label plate reader, while the release of IL12 (**f**) in tissue culture supernatants was measured by ELISA. Graphs in d-e show data from at least three independent experiments.

**Figure 4 biomedicines-12-01577-f004:**
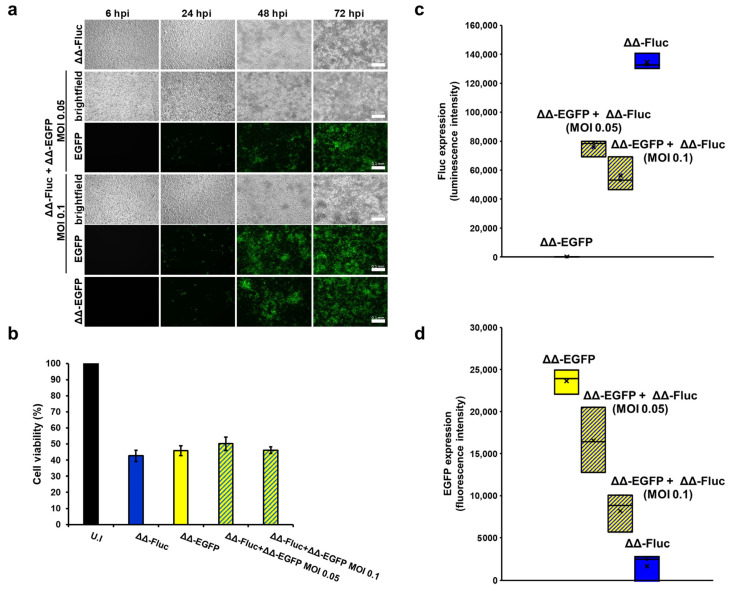
MDA-MB-231 cells infected with a combination of ΔΔ-EGFP and ΔΔ-Fluc express both transgenes and show clear virus-induced cytopathic effects. MDA-MB-231 cells were co-infected with ΔΔ-EGFP and ΔΔ-Fluc combined at an MOI of 0.1 or 0.05 PFU/cell each. As a control, cells were also infected with single recombinant viruses (MOI 0.1 PFU/cell). (**a**) At the indicated time post infection, cells were examined with a Leica epifluorescence DC100 microscope. Representative images are shown. Scale bars: 0.1 mm. (**b**) At 72 hpi, cells were harvested and cell viability was determined by trypan blue staining. The graphs display the mean values of the percentage of live cells compared to uninfected cells from three independent experiments. Error bars represent the SE. At the same time, the intensity of Fluc (**c**) and EGFP (**d**) signals was measured from infected cells. Graphs in c and d show data from three independent experiments.

**Figure 5 biomedicines-12-01577-f005:**
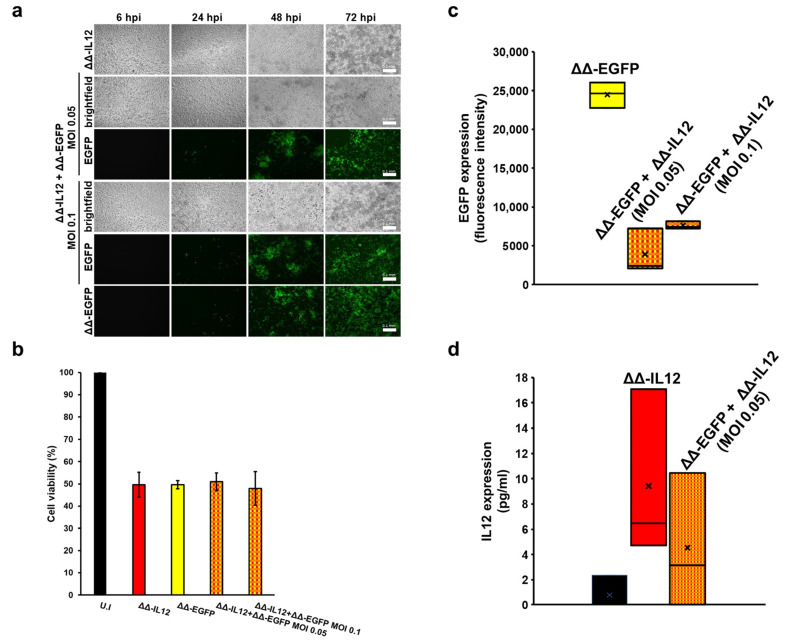
MDA-MB-231 cells infected with a combination of ΔΔ-EGFP HSV-1 and ΔΔ-IL12 express both transgenes and show clear virus-induced cytopathic effects. MDA-MB-231 cells were co-infected with ΔΔ-EGFP and ΔΔ-IL12 combined at an MOI of 0.1 or 0.05 PFU/cell each. As a control, cells were also infected with single recombinant viruses (MOI 0.1 PFU/cell). (**a**) At the indicated times post infection, cells were observed by fluorescence microscopy. Scale bars: 0.1 mm. Representative images are shown. (**b**) At 72 hpi, cells were harvested and cell viability determined by trypan blue staining. The mean values of the percentage of live cells as compared to uninfected cells from three independent experiments are shown. Error bars represent the SE. At the same time pi, the intensity of the EGFP signal (**c**) and the amounts of IL12 (pg/mL) released in supernatants of infected MDA-MB-231 (**d**) were measured. Graphs in c and d show data from three independent experiments.

**Figure 6 biomedicines-12-01577-f006:**
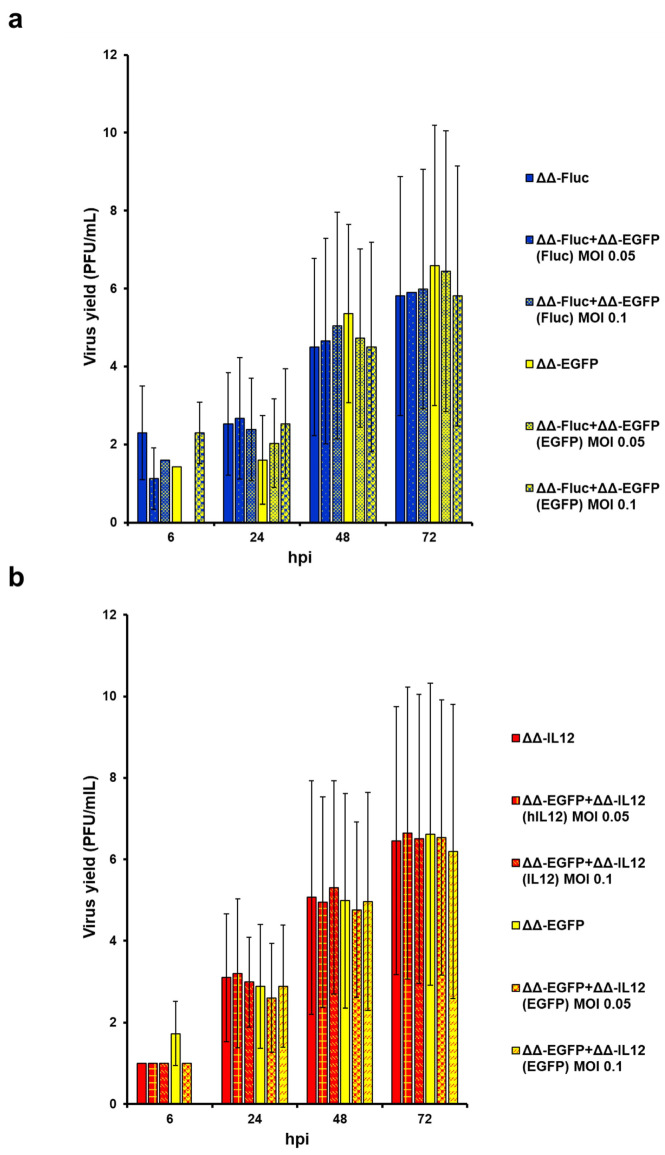
Recombinant viruses used in combination replicate in MDA-MB-231 as efficiently as in a single infection. MDA-MB 231 cells were co-infected with either ΔΔ-EGFP and ΔΔ-Fluc (**a**) or with ΔΔ-EGFP and ΔΔ-hIL12 (**b**) at the indicated MOIs. Single viruses were also adopted for infection as a control at an MOI of 0.1 PFU/cell. At the indicated hpi, cell supernatants were harvested and a plaque assay was performed in Vero cells to evaluate viral titers expressed in PFU/mL. The graphs show the mean and standard errors (SEs) from three independent experiments. Y axis is a logarithmic scale.

## Data Availability

The original data presented in the study are openly available in Research data UNIPD at https://researchdata.cab.unipd.it/id/eprint/1319, accessed on 4 July 2024, https://researchdata.cab.unipd.it/id/eprint/1320 accessed on 4 July 2024 and https://researchdata.cab.unipd.it/id/eprint/1321, accessed on 4 July 2024.

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
