# Peer review of "Simultaneous Expression of Different Therapeutic Genes by Infection with Multiple Oncolytic HSV-1 Vectors"

_biomedicines, 2024, doi:10.3390/biomedicines12071577_

Round 1

Reviewer 1 Report

Comments and Suggestions for Authors

Comment:

In this study, the authors investigate the use of several oncolytic herpes simplex virus type 1 (oHSV-1) vectors for the simultaneous expression of different therapeutic genes in triple-negative breast cancer (TNBC) cells. It shows that co-infection with oHSV-1, each encoding a single transgene, leads to effective expression of multiple proteins while maintaining comparable efficacy in killing cancer cells and viral replication. The entire work was well organized and could be published after addressing some important concerns:

1. How did the authors ensure that the viral constructs were stable under all conditions?

2. Why did the authors choose MDA-MB-231 cells for the expression of different transgenes from viruses? Are other cell types such as CHO or VERO cells equally efficient? The authors are advised to repeat these experiments in at least two cell types.

3. Are any off-target effects of the recombinant viruses on non-cancer cells observed? The authors are advised to confirm the biological safety.

4. In addition, how were the viral titers optimized in this study?

5. The authors are advised to add more detailed descriptions of the potential immune response induced by the oncolytic viruses that could impact subsequent cancer treatments or cancer recurrence in this manuscript.

Comments on the Quality of English Language

Need to be improved.

Author Response

Comment 1: In this study, the authors investigate the use of several oncolytic herpes simplex virus type 1 (oHSV-1) vectors for the simultaneous expression of different therapeutic genes in triple-negative breast cancer (TNBC) cells. It shows that co-infection with oHSV-1, each encoding a single transgene, leads to effective expression of multiple proteins while maintaining comparable efficacy in killing cancer cells and viral replication. The entire work was well organized and could be published after addressing some important concerns.

Response 1: We would like to thank Reviewer 1 for positively considering our study and her/his constructive criticisms. Here below, there is a point-by-point response to her/his questions.

Comment 2: How did the authors ensure that the viral constructs were stable under all conditions?

Response 2: We thank the reviewer for raising this important point. We have considered this and would like to point out that the viral constructs generated and used in this study are based on previous work carried out and published by various research groups. First, all recombinant viruses derive from a bacmid (reported as BAC-Δγ34.5 in the manuscript and described at pg. 3, lines 105-110), which contains the genome of HSV-1 strain 17syn+, with a deletion of both loci of the γ34.5 gene. Furthermore, this bacmid already contains an expression cassette for the Firefly luciferase (Fluc) under the control of the HCMV immediate-early promoter, which was inserted in the intergenic region of open reading frames UL55 and UL56. This bacmid was developed by Prof. Beate Sodeik (University of Hannover) and it was already used, for instance, to generate an oncolytic HSV-1 for the treatment of the experimental autoimmune encephalomyelitis (EAE) (Nygårdas M, et al, PLoS One. 2013;8(5):e64200, reported as ref. #25 in our manuscript). In this paper, Nygårdas and coworkers analyzed viral spread in vivo by detecting “HSV DNA by PCR in brain, trigeminal ganglion and spinal cord samples during the acute infection (Table S2)”, as well as the transgene sequence on day 9 post EAE induction, “but also in mice on day 14 and day 21 (Table S2)”. The Authors also show that Fluc is expressed in vivo (Figure 3A from Nygårdas et al.), demonstrating transgene expression from this locus by the HCMV promoter in an in vivo model in various tissues. Since this locus seems to tolerate insertions of sequences without destabilizing the viral genome and allows for continuous transgene expression, we decided to use the same locus for the insertion of the EGFP and the IL12 encoding sequences in our constructs by keeping the HCMV promoter driving the transgene expression.

Before the substitution of the transgenes, we performed mutagenesis on BAC-Δγ34.5 to achieve the same Us12 deletion described in T-VEC [Liu BL, et al., Gene Ther. 2003;10:292–303, ref. #19 in our manuscript], generating ΔΔ-Fluc. The bacmid of ΔΔ-Fluc was used then for the substitution of transgeneses. T-VEC has been widely used in vitro and in vivo [see for instance the review reported as ref. #5 in our manuscript] and has to our knowledge no reported stability issues. Furthermore, the in this study used ΔΔ-EGFP virus was already used in a previous study, where we did not notice issues related to its stability, both in vitro and in vivo [Reale A et al., Int J Mol Sci 2023;24, ref # 26 of our manuscript]. Although there was no reason to doubt the stability of our constructs based on the data, they were still checked during the generation of virus stocks and in the experiments. In fact, all stocks viruses were controlled for transgene expression prior to their use in experiments. Although we cannot completely exclude unintended changes, since we did not sequence the viral genomes after virus reconstitution, we can say that all recombinant viruses used in the work did not show any alteration in their replication, plaque formation and transgene expression. Nevertheless, we addressed the issue of stability experimentally by infecting Vero cells with our stock viruses at a MOI of 0.01 PFU/cell. The culture supernatant collected after 72 hours of incubation was then used to pass the virus on fresh Vero cells. This procedure was repeated a total of 8 times over a period of 28 days. At the end, the cells were then examined under the microscope to check for the presence of cytopathic effects and the expression of the transgene, if this was easily possible (e.g. in the case of EGFP). Again, we did not find any signs of instability of our viruses.

We feel that we have sufficiently controlled the stability of our viruses for this study. Since these are important controls, we have added a note to the revised manuscript describing our experiments to test virus stability (at pg 4, lines 158-166). In addition, we have added a more detailed description of our viral constructs and the methods used to check the viral stocks in the revised version of the manuscript in the section “Materials and Methods” (pgs. 3 and 4, lines 110-157; pg. 4, lines 186-192). We also added a comment on the stability in the section “Conclusions” (pg. 13, line 487)

Comment 3: Why did the authors choose MDA-MB-231 cells for the expression of different transgenes from viruses? Are other cell types such as CHO or VERO cells equally efficient? The authors are advised to repeat these experiments in at least two cell types

Response 3: We thank the reviewer for the question. It was our intention to provide a rationale for the use of MDA-MB-231 cells with the following statement in the manuscript “In search for oHSV-1 candidates against human breast cancer, we evaluated the replication capacity of a Δγ34.5 HSV-1 in the MDA-MB-231 cell line, a widely accepted cell culture model for triple-negative breast cancer (TNBC) [28]. TNBC is an aggressive tumor that responds poorly to current treatments and for which oncolytic viruses (OVs) may represent a therapeutic option, particularly in synergy with immunotherapy.” As we understand that these points were not sufficiently emphasized, we hope to have clarified this in the revised version of the manuscript (p. 5, lines 210-213; p. 13, lines 425-430; new refs #38, 39, 40). The MDA-MB-231 cell line is widely accepted and used as an in vitro model for experimental work on triple negative breast cancer [(TBNC); see for instance ref #30 of the revised version of the manuscript]. TBNC is a perfect target for the application of a combinatorial approach of viroimmunotherapy (see for instance new refs #38, 39, 40 in the revised version of the manuscript), as the one we are proposing in our study. Therefore, we decided to use MDA-MB-231 cells for this in vitro proof-of-principle study to investigate the feasibility of a mixed infections-based strategy for simultaneous expression of different transgenes in cancer cells.

We also understand the importance of confirming the feasibility of our strategy in cells other than MDA-MB-231, as requested by the Referee, and have therefore included in the revised manuscript results of an experiment in which Vero cells were infected with supernatants from mono and co-infected MDA-MB-231 cells. We were able to show that expression of both EGFP and Fluc was still detectable under all tested conditions (Figure S3 of the revised version of the manuscript). These results demonstrate that in mixed infections of MDA-MB-231 cells, viral particles are produced that as lead to the expression of both transgenes in Vero cells.  

We would also like to point out that we have already been able to show that the transgenes of our viral constructs are also expressed in cancer cells other than MDA-MB-231 [Reale A et al., Int J Mol Sci 2023;24., ref # 26 of our manuscript]. For example, we were able to show that infection of human acute monocytic leukemia cell line (THP-1 cells) and human head and neck squamous cell carcinoma (UM-SCC-11B) cells with ΔΔ-EGFP leads to the expression of EGFP in these cells.

Comment 4: Are any off-target effects of the recombinant viruses on non-cancer cells observed? The authors are advised to confirm the biological safety.

Response 4: Although we understand the concern of the reviewer, we would like to point out that we have conducted an in vitro study. The more relevant approach to address safety issues is to use an in vivo model, which we plan to do in future experiments. Furthermore, our constructs have the same genetic modifications as T-VEC. T-VEC has been extensively tested for safety in both animal models and clinical trials and is already approved for the treatment of patients with advanced melanoma. In addition, it is currently being tested in clinical trials, together with other oncolytic viruses, for the treatment of other tumor types and has not shown any serious side effects so far [see for instance data reported in refs #9 and 18 of the revised version of the manuscript]. This virus, like most of the oncolytic viruses, is attenuated and engineered to limit its replication mainly to cancer cells. Furthermore, our constructs, like T-VEC, still express the viral thymidine kinase (TK), which is an additional safety measure. Therefore, acyclovir and its derivatives can be used to treat patients should side effects occur during treatment with the oHSV-1. The strategy of combinatorial infection with two oncolytic HSV-1 (one expressing a mouse anti-PD-1 antibody, the other one encoding for the murine IL-12) for the treatment of colon adenocarcinoma has already been tested in vivo in a mouse model by Xie and coworkers [ref. #37 of our manuscript]. This study not only showed inhibition of the primary tumor growth, but also the safety of this approach with a significant increase in overall survival of the mice.

However, to give more emphasis to this important aspect, we have expanded the relevant sections on safety concerns in the discussion and conclusion of the revised manuscript (pg. 1, line 38; pg. 12, lines 386-388; pg. 14, lines 485-488) and added additional references (refs. # 4, 5, 9, 17). In addition, we have further emphasized the concept of T-VEC attenuation and safety in the introduction of the manuscript (pg. 2, lines 60-64).

Comment 5:  In addition, how were the viral titers optimized in this study?

Response 5. We thank the reviewer for this comment. We probably did not describe this experiment sufficiently in the previous version. We have now added a new paragraph to the “Materials and Methods” section of the manuscript that describes the virus titration (pgs. 2-3, lines 128-166). In the paragraph entitled “Immunofluorescence assay” we have also clarified that in all experiments comparing different viruses, e.g. with regard to transgene expression, virus spread, and killing activity, the same infection of MDA-MB-231 cells was controlled by determining the HSV-1 immediate early protein ICP4 at an early time post infection (i.e. 6h) (pg. 4, lines 186-192). This means that, in addition to using the same MOI based on the stock virus titers, we also determined the actual infection rate and thus ensured the same infection in our experiments.

Comment 6: The authors are advised to add more detailed descriptions of the potential immune response induced by the oncolytic viruses that could impact subsequent cancer treatments or cancer recurrence in this manuscript.

Response 6. We thank the reviewer for this helpful suggestion. We have included a more in depth description regards the impact of oncolytic viruses in general and of T-VEC in particular, on the immune response against cancer and on its impact of the therapy success (pg. 1, lines 65-66; pg. 12, lines 372-385; pg. 14, lines 485-488). In addition, we emphasized that Xie and co-workers [ref. #37 of our manuscript] were able to demonstrate the induction of a vaccine-like anti-tumor response when two oncolytic HSV-1 expressing immunotherapeutic factors were administered simultaneously, and how this might impact treatment in human patients (pg. 12, lines 410-418). Finally, we have explained the role of IL12 in this context in more detail (pg. 6, lines 264-265 and pg. 12, lines 416-418).

Reviewer 2 Report

Comments and Suggestions for Authors

The manuscript “Simultaneous expression of various therapeutic genes by infection with multiple oncolytic vectors of HSV-1” presents an interesting approach to virotherapy of oncological diseases: the simultaneous introduction of several variants of the same virus carrying different transgenes. The work is well founded and thought out.

Using a complex of genetic engineering and virology methods, the authors obtained variants of the herpes simplex virus carrying a reporter protein or human interleukin 12. The positive result of genetic modification of the herpes virus was demonstrated on the MDA-MB-231 cell line, which was adopted as a model of a cell culture of triple negative breast cancer. 

The authors, first of all, showed that simultaneous infection of cells with several variants of the herpes virus does not affect their reproduction, the level of which determines the direct oncolytic effect of the virus. This result is fundamentally important for assessing the effectiveness of genetically modified viruses on cancer cells.

The experimental part of the work is convincingly illustrated with microphotographs and graphs; the results obtained are beyond doubt. In the “Discussion” section, the authors carefully analyze the results obtained not only in a positive aspect, but also in terms of possible shortcomings of the developed approach, which will help readers who are not familiar with the intricacies of research on oncolytic viruses to understand and value this study.

Thus, the present study showed that infection of MDA-MB-231 cells with a combination of ΔΔ-EGFP HSV-1 and ΔΔ-IL12 ensures the expression of both transgenes and is accompanied by pronounced cytopathic effects due to the reproduction of the herpes virus.

The work presented in the article “Simultaneous expression of various therapeutic genes during infection with multiple oncolytic vectors of HSV-1” is important for the development of tumor virotherapy; The results obtained open a new field for research and development of new viral drugs that do not have the known disadvantages of virotherapy. Undoubtedly, the publication of the article will arouse great interest among researchers working in this area. I believe that this article can be published in the Biomedicines in its current form.

Author Response

Comment 1: The manuscript “Simultaneous expression of various therapeutic genes by infection with multiple oncolytic vectors of HSV-1” presents an interesting approach to virotherapy of oncological diseases: the simultaneous introduction of several variants of the same virus carrying different transgenes. The work is well founded and thought out. Using a complex of genetic engineering and virology methods, the authors obtained variants of the herpes simplex virus carrying a reporter protein or human interleukin 12. The positive result of genetic modification of the herpes virus was demonstrated on the MDA-MB-231 cell line, which was adopted as a model of a cell culture of triple negative breast cancer.  The authors, first of all, showed that simultaneous infection of cells with several variants of the herpes virus does not affect their reproduction, the level of which determines the direct oncolytic effect of the virus. This result is fundamentally important for assessing the effectiveness of genetically modified viruses on cancer cells. The experimental part of the work is convincingly illustrated with microphotographs and graphs; the results obtained are beyond doubt. In the “Discussion” section, the authors carefully analyze the results obtained not only in a positive aspect, but also in terms of possible shortcomings of the developed approach, which will help readers who are not familiar with the intricacies of research on oncolytic viruses to understand and value this study. Thus, the present study showed that infection of MDA-MB-231 cells with a combination of ΔΔ-EGFP HSV-1 and ΔΔ-IL12 ensures the expression of both transgenes and is accompanied by pronounced cytopathic effects due to the reproduction of the herpes virus. The work presented in the article “Simultaneous expression of various therapeutic genes during infection with multiple oncolytic vectors of HSV-1” is important for the development of tumor virotherapy; The results obtained open a new field for research and development of new viral drugs that do not have the known disadvantages of virotherapy. Undoubtedly, the publication of the article will arouse great interest among researchers working in this area. I believe that this article can be published in the Biomedicines in its current form.

Response1 : We would like to thank Reviewer 2 for positive assessment of our study and for pointing out its strengths.

Round 2

Reviewer 1 Report

Comments and Suggestions for Authors

The manuscript is recommended to be accepted now.

Comments on the Quality of English Language

It's good now.